# Customizing EV-CATCHER to Purify Placental Extracellular Vesicles from Maternal Plasma to Detect Placental Pathologies

**DOI:** 10.3390/ijms25105102

**Published:** 2024-05-07

**Authors:** Megan I. Mitchell, Marwa Khalil, Iddo Z. Ben-Dov, Jesus Alverez-Perez, Nicholas P. Illsley, Stacy Zamudio, Abdulla Al-Khan, Olivier Loudig

**Affiliations:** 1Center for Discovery and Innovation, Hackensack Meridian Health, Nutley, NJ 07110, USA; megan.mitchell@hmh-cdi.org; 2Hackensack University Medical Center, Department of Pediatrics, Hackensack Meridian Health, Hackensack, NJ 07601, USA; marwa.khalil@hmhn.org; 3Hackensack Meridian School of Medicine (HMHSOM), Nutley, NJ 07110, USA; jesus.alverez-perez@hmhn.org (J.A.-P.); abdulla.al-khan@hmhn.org (A.A.-K.); 4Laboratory of Medical Transcriptomics, Hadassah-Hebrew University Medical Center, Jerusalem 91120, Israel; iddobe@ekmd.huji.ac.il; 5Hackensack University Medical Center, Department of Maternal and Fetal Medicine, Hackensack Meridian Health, Hackensack, NJ 07601, USA; placresgp@gmail.com (N.P.I.);

**Keywords:** EV-CATCHER, extracellular vesicles, placenta, PAS, percreta, previa, NGS, microRNA, ONi, PLAP, placental alkaline phosphatase, abnormal placentation, biomarker

## Abstract

Placenta Accreta Spectrum (PAS) is a life-threatening condition in which placental trophoblastic cells abnormally invade the uterus, often up to the uterine serosa and, in extreme cases, tissues beyond the uterine wall. Currently, there is no clinical assay for the non-invasive detection of PAS, and only ultrasound and MRI can be used for its diagnosis. Considering the subjectivity of visual assessment, the detection of PAS necessitates a high degree of expertise and, in some instances, can lead to its misdiagnosis. In clinical practice, up to 50% of pregnancies with PAS remain undiagnosed until delivery, and it is associated with increased risk of morbidity/mortality. Although many studies have evaluated the potential of fetal biomarkers circulating in maternal blood, very few studies have evaluated the potential of circulating placental extracellular vesicles (EVs) and their miRNA contents for molecular detection of PAS. Thus, to purify placental EVs from maternal blood, we customized our robust ultra-sensitive immuno-purification assay, termed EV-CATCHER, with a monoclonal antibody targeting the membrane Placental Alkaline Phosphatase (PLAP) protein, which is unique to the placenta and present on the surface of placental EVs. Then, as a pilot evaluation, we compared the miRNA expression profiles of placental EVs purified from the maternal plasma of women diagnosed with placenta previa (controls, *n* = 16); placenta lying low in uterus but not invasive) to those of placental EVs purified from the plasma of women with placenta percreta (cases, *n* = 16), PAS with the highest level of invasiveness. Our analyses reveal that miRNA profiling of PLAP^+^ EVs purified from maternal plasma identified 40 differentially expressed miRNAs when comparing these two placental pathologies. Preliminary miRNA pathway enrichment and gene ontology analysis of the top 14 upregulated and top nine downregulated miRNAs in PLAP^+^ EVs, purified from the plasma of women diagnosed with placenta percreta versus those diagnosed with placenta previa, suggests a potential role in control of cellular invasion and motility that will require further investigation.

## 1. Introduction

Placenta accreta spectrum (PAS), also referred to as morbidly adherent placenta, denotes the abnormal adherence and invasion of the placental trophoblast into the uterine myometrium [1,2,3,4]. PAS is pathologically classified into three different categories depending on the depth of trophoblast invasion (i.e., accreta, increta and percreta). Specifically, placenta accreta refers to the direct attachment of the placenta to the myometrium, while placenta increta refers to the invasion of the trophoblast to a greater than normal depth into the myometrium. Placenta percreta, on the other hand, is considered the most severe form of PAS, leading to the invasion of the trophoblast through the myometrium, up to the uterine serosa (and frequently beyond), and then into the surrounding tissues and organs [5]. Although the pathogenesis of PAS remains unclear, its pathophysiology is associated with two major risk factors. The first risk factor is uterine scarring from previous surgical (e.g., Cesarean delivery) or other uterine trauma, which enables the trophoblast to attach to and invade the scarred myometrium [6,7,8,9]. The second risk factor is placenta previa, which is the implantation of the placenta over the cervical ostium (os). It is thought that the thinned endometrial layer around the os provides a lower degree of resistance to trophoblast invasion. The degree of abnormal placentation is correlated with severe clinical implications, as the failure of the placenta to spontaneously detach from the uterus during delivery is associated with significant risk of major maternal hemorrhage. This can result in disseminated intravascular coagulation (DIC), multisystem organ failure, and, in extreme cases, death [10,11,12,13,14,15,16].

The incidence of placenta accreta spectrum (PAS) has increased and has been reported to be as high as 1–2 in 1000 births in tertiary care settings [17,18,19]. These conditions have become one of the leading causes of postpartum hemorrhage in the U.S., remaining a significant contributor to maternal morbidity and mortality. Although the risk factors of prior uterine incision (i.e., primarily from previous Cesarean sections) is of major epidemiological importance for identifying patients at risk, there are nevertheless cases in which the presence of this risk factor is not associated with PAS. In clinical practice, up to 50% of pregnancies with PAS remain undiagnosed until delivery, and thus they are associated with an increased risk of morbidity [6,7]. Considering that there are currently no clinical diagnostic assays routinely used to detect the development of PAS, new and improved paradigms are urgently needed for the early and accurate diagnosis of this condition. Although both ultrasound and MRI have been used effectively to diagnose certain PAS cases, the subjectivity involved in assessing visual markers remains restricted to expert use and trained professionals [20,21,22]. The consequence is that many cases of PAS remain undiagnosed or misdiagnosed, leading to poor maternal outcomes. In view of these circumstances, having biomarkers for the detection of PAS would be of considerable diagnostic benefit, enabling physicians to prepare early enough for the complex delivery often required in these cases [23,24,25,26,27]. Additionally, the identification of the molecular pathways involved in the upregulation of placental tissue invasion might enable the design of novel therapeutic tools to help reduce the degree of invasion and, in turn, regulate the incidence of these abnormal placental conditions.

To date, many biomarker studies have been conducted, but many have been inconclusive, and there are still no clinically reliable urine or blood biomarkers for the early detection of PAS. Although several studies suggest that impaired angiogenesis [28,29,30], abnormal decidualization [31], and trophoblast factors [32,33] contribute to the pathophysiology of PAS, evaluations of the maternal serum for angiogenic and aneuploidy markers, as well as fetal circulating DNA, obtained during non-invasive prenatal screenings, have not identified robust biomarkers, which could provide a clinically useful diagnostic test for PAS [34,35,36,37,38,39,40,41,42,43,44,45,46,47,48,49]. Nevertheless, over the last several years, many studies have been focused on the identification of fetal biomarkers circulating in maternal blood, which is in direct contact with the placenta [18,19,39]. Several placental and fetal hormones routinely used in the screening for aneuploidy have been found to be differentially expressed in the serum of women with PAS compared with those with placenta previa [44,45]. Recently, there has been increasing interest in the role of cell-free fetal DNA (cffDNA) encapsulated within extracellular vesicles (EVs) for the screening and diagnosis of PAS, but these studies are still ongoing [37,38].

The term extracellular vesicle (EV) describes all small bi-layered membrane-bound vesicles including nanosomes (8–12 nm), exosomes (30–150 nm), microvesicles (MVs) (200–1000 nm), and apoptotic bodies (500+ nm) [50]. The formation of microvesicles and nanovesicles is believed to occur through the outward blebbing and detachment of the plasma membrane in a controlled process that may utilize endosomal machinery [51]. Exosome formation, on the other hand, is a distinct process which originates from the inward budding of the plasma membrane and the subsequent formation of multivesicular bodies (MVBs) through the endosomal pathway, which results in the extracellular release of exosomes upon the fusion of MVBs with the plasma membrane [52]. Recently, the International Society for Extracellular Vesicles (ISEV) published an updated version of their minimal information for studies of extracellular vesicles (MISEV2023) guidelines [53] in which they endorse the utilization of the term “extracellular vesicle” (EV) for any naturally released particle with a lipid membrane bilayer, which cannot self-replicate. In accordance with these guidelines, we utilize the term EV for all descriptions in this manuscript.

Placental cells have been shown to shed EVs into both fetal and maternal circulations that can be detected as early as the 6th week of pregnancy [54,55]. The abundance of placental EVs circulating in maternal blood increases in amount throughout the duration of pregnancy, with maximal levels being detected at term [56]. As such, these circulating placental EVs represent a potentially valuable source of placenta-specific biomarkers for the non-invasive diagnosis of PAS [57]. One of the defining characteristics of placental EVs is the presence of placental alkaline phosphatase (PLAP), a membrane protein uniquely produced by the placenta [58,59,60]. Although other alkaline phosphatase proteins can be found in all human tissues, PLAP lacks the last 24 amino acids of its N-terminal region, which makes it unique and specific to the placenta, providing distinct epitopes for its targeted antibody capture. This N-terminal modification increases substrate specificity, as well as stability to heat and resistance to chemical inactivation. To date, the main functions of PLAP that have been described include assistance in the transfer of immunoglobulin G (IgG) from the mother to the fetus and the stimulation of fibroblast DNA synthesis and proliferation [54]. As PLAP is known to be a membrane protein found in abundance on placental EVs, this unique surface marker has become a target for the purification of placental EVs [61]. Recent studies conducted on placental EVs purified from maternal circulation have already demonstrated that they can participate in the adaptive immune response in the mother and fetus, and also that their concentration and function differ in various placental pathologies [62,63,64,65].

For this study, we utilized our state-of-the-art EV antibody-based purification assay (EV-CATCHER) to selectively isolate placental EVs from maternal plasma; we previously demonstrated its ability to capture ultra-pure EVs, when compared to commercially available assays, and that increases signal-to-noise ratio for miRNA analyses [66]. We customized our EV-CATCHER assay with an anti-PLAP antibody and then used our optimized small-RNA cDNA library preparation protocol to conduct NGS analyses [66] where we compared the miRNA profiles of circulating placental EVs purified from the plasma of pregnancies with placenta percreta (cases) with those purified from plasma of placenta previa pregnancies (controls). We specifically selected placenta previa pregnancies as our controls to avoid the confounding effects of gestational age, mode of delivery (i.e., non-laboring Cesarean vs. laboring with vaginal or Cesarean delivery), and uterine localization of the placenta in order to provide greater rigor in the identification of EV-miRNA signatures uniquely associated with PAS [11].

## 2. Results

### 2.1. Validation of PLAP Customization of EV-CATCHER

In order to validate our PLAP customized EV-CATCHER assay for the isolation of placental EVs from maternal blood, we first tested our monoclonal PLAP antibody for its ability to recognize the recombinant placental alkaline phosphatase (PLAP), compared to a recombinant ubiquitously expressed tissue-nonspecific isozyme of alkaline phosphatase (ALPL) protein, using Western blot analyses. We demonstrated that our PLAP antibody reacted strongly with the recombinant PLAP protein with no cross-reactivity against the recombinant ALPL protein (Figure 1a). Then, we sought to characterize the EVs isolated with our monoclonal PLAP antibody customized EV-CATCHER assay (Figure 1b), and thus used transmission electron microscopy (TEM) imaging and ONi super resolution nanoimaging to evaluate those purified from the maternal plasma of a woman with placenta previa and a woman with placenta percreta. Both TEM (Figure 1c) and ONi super resolution nanoimaging (Figure 1d) analyses demonstrated that the morphology and size of PLAP^+^ (positive) EVs isolated from maternal plasma are consistent with those of EVs that we previously purified from human plasma with our EV-CATCHER assay [66] and what is currently described in the literature for extracellular vesicles isolated from plasma [67]. Moreover, our ONi super resolution nanoimaging analyses, which specifically evaluated the localization of CD9, CD63, and PLAP proteins on the surface of immobilized EVs, isolated from the plasma of women with previa and percreta pregnancies using our PLAP customized EV-CATCHER assay, demonstrated equal distribution levels of these three surface markers between the two different plasma samples (Figure 1d, large field-of-view images). We observed that the EV-specific CD9 tetraspanin protein (Figure 1d, green fluorescent signal on left panels (i.e., large field-of-view) for previa and percreta plasma samples) was the most abundantly detected protein on the surface of immobilized EVs, which appeared similarly detected on the surface of EVs purified from our two different plasma samples (i.e., previa and percreta). We observed that the EV-specific CD63 tetraspanin was also detectable on most of the immobilized EVs, in addition to being similarly detected between the two different plasma samples, although it appeared less abundantly expressed on these EVs than CD9 (Figure 1d, yellow fluorescent signal on left panels (i.e., large field-of-view) for previa and percreta plasma samples). Finally, we confirmed the presence of PLAP proteins on the surface of immobilized EVs, which we revealed to be the least abundant of all three proteins, as observed by the lower number of protein clusters in the large field-of-view (Figure 1d, red fluorescent signal on left panels for previa and percreta plasma samples) when compared to CD9 and CD63. Although not observable on the large field-of-view, when we focused our imaging on single EVs, we observed the presence of PLAP on EVs not distinguishable in the large field-of-view. Thus, we provide images that allowed us to identify high PLAP expressing EVs (i.e., PLAP protein clusters) and low-PLAP expressing EVs (i.e., single PLAP proteins) that could be detected for both previa and percreta plasma samples (Figure 1d, right panels for previa and percreta plasma samples).

### 2.2. miRNA Analysis of PLAP^+^ EVs Purified from Maternal Plasma of Previa and Percreta Pregnancies Using the PLAP Customized EV-CATCHER Assay

We sought to determine whether the PLAP antibody customization of our ultra-sensitive EV-CATCHER purification assay [66] allowed for detection of differentially expressed miRNAs between PLAP^+^ EVs isolated from placenta previa and placenta percreta maternal plasma samples. Therefore, using RNA extracted from PLAP^+^ EVs isolated with EV-CATCHER from the plasma of women diagnosed with placenta previa (*n* = 16; controls) and women diagnosed with placenta percreta (cases; *n* = 16), with a mean gestational age of 35 ± 4 weeks and 33 ± 4 weeks, respectively (Figure 2), we prepared small-RNA cDNA libraries for next generation sequencing (NGS) analyses. Demographic information for all subjects is displayed in Appendix A. We prepared 2 cDNA libraries that included the 32 samples, with each library containing an even number of cases and controls, to minimize bias and batch effect. Heatmap evaluations of our NGS data demonstrated that we were able to differentiate almost perfectly between EVs isolated from the plasma of placenta previa and placenta percreta pregnancies based on the differential expression of miRNAs and without supervised clustering between the two pathologies (Figure 3a). Upon further analyses, we identified a total of 40 significantly (*p*-value < 0.05) differentially expressed EV-miRNAs between placenta previa and percreta sample groups (Figure 3b). When we established an integrative miRNA signature, using the 40 differentially expressed miRNAs extracted from PLAP^+^ EVs, we observed an even greater discrimination between the two sample groups, with a *p*-value estimated at 7.1 × 10^−5^ (Figure 3c). Interestingly, the global expression trend for miRNAs in EVs purified from plasma of placenta percreta pregnancies indicated an overall downregulation (Figure 3c). This decrease in overall miRNA expression is biologically compatible with a decrease in mRNA target inhibition and a potential increase in cellular proliferation and invasion. Analysis of the differentially expressed miRNAs with the highest NGS read counts (i.e., baseMean > 1000), identified six significant differentially expressed miRNAs (*p* < 0.05) between previa and percreta sample groups (Figure 4; hsa-miR-486 (*p*-value = 0.031), hsa-miR-151-3p (*p*-value = 0.017), hsa-miR-378 (*p*-value = 0.007), hsa-miR-122 (*p*-value = 0.001), hsa-miR-199a-5p (*p*-value = 0.027), and hsa-miR-340 (*p*-value = 0.014), all of which have been described to be associated with oncogenic invasive and proliferative processes, as addressed in the discussion.

### 2.3. miRNA Pathway Enrichment Analysis and Prediction of Biological Pathway Involvement Using Gene Ontology (GO)

Since miRNAs directly bind mRNA targets and thus regulate gene expression through mRNA destabilization and inhibition of gene translation [68], it has been shown that, for any list of differentially expressed miRNAs, the biological processes they post-transcriptionally regulate may be predicted using in silico enrichment analysis tools. Thus, using miRNA enrichment analysis (miRNet), we conducted preliminary miRNA pathway enrichment analyses, employing the top 14 upregulated and top nine downregulated EV-miRNAs that were detected in placental EVs circulating in the plasma of placenta percreta pregnancies in comparison with placenta previa pregnancies. Our enrichment analyses determined that the 14 upregulated miRNAs were predicted to regulate gene targets, including AKT1, IGFR1, TP53, PIK3C2A, ZEB1 and FOXO1. These are known to be involved in cell migration, cell proliferation, and angiogenesis (Figure 5a). Assessment of the top 9 miRNAs, which expression appeared downregulated in placental EVs purified from the plasma of placenta percreta pregnancies, identified predicted gene interactions with KRAS, GSK3β and CCND1, which are genes known to play regulatory effects in cell proliferation, migration, and sprouting angiogenesis (Figure 5b).

## 3. Discussion

In this pilot study, we isolated, analyzed, and profiled the miRNA cargos of placental (PLAP^+^) EVs isolated from the maternal plasma of women with placenta previa (control) and placenta percreta (case) pregnancies using an antibody targeting the placental alkaline phosphatase (PLAP) membrane protein, which is uniquely produced by placental cells [61]. Using next generation sequencing data analysis, we assessed the potential miRNA expression differences in circulating placental EVs that may contribute to the diagnosis of PAS. Our study, which adheres to current clinical and scientific guidelines, used maternal plasma specimens where histopathologic confirmation of case diagnosis and differentiation between PAS grades was confirmed prior to our analyses. We used the FIGO Clinical Classification System to allow for multi-center standardization. The decision to utilize patients diagnosed with placenta previa as controls for this study was guided by the fact that the presence of placenta previa alone, regardless of uterine scarring, increases PAS risk by 100-fold, as previously described [69,70]. Indeed, implantation of the placenta over the cervical ostium (i.e., placenta previa), combined with Cesarean scarring, increases the underlying risk of PAS by >10%, and each subsequent Cesarean section more than doubles the risk of the development of PAS [2,69]. Furthermore, by anticipating comparable gestational age effects with placenta percreta, these controls allowed our analyses to be focused on EV-miRNA signatures associated with the invasive phenotype of PAS.

Prior to conducting our miRNA analyses, we validated the selectivity of our PLAP antibody customized EV-CATCHER assay for the purification of PLAP^+^ EVs. We used TEM to confirm the size and morphology and ONi super-resolution nanoimaging to confirm the distribution of the EV-specific tetraspanins CD9 and CD63, which appeared to be similarly detected between EVs isolated from both the plasma of previa and percreta pregnancies. Using ONi, we also confirmed the presence of the PLAP proteins on the surface of these EVs. However, we observed that PLAP proteins were generally lower in the number of copies on the surface of individual EVs than CD9, which appeared to be in large clusters on these EVs. In the large field-of-view images, we observed high-PLAP-expressing EVs (i.e., >50 PLAP copies), but could not distinguish the low-PLAP-expressing EVs (i.e., <5 PLAP copies). Therefore, we focused our nanoimaging onto individually immobilized EVs, which revealed that most EVs harbored PLAP proteins and that only a few EVs did not display any PLAP proteins on their surface. This could be explained by a lack of access to the PLAP epitope on the surface of these EVs, which may be due to a blockage of the epitope by the PLAP purifying antibody (i.e., released with the EVs after EV-CATCHER purification, or a lack of physical access to PLAP proteins localized on the side of the EV that was immobilized onto the ONi platform). It is possible that some EVs could have been purified non-specifically, but this is very unlikely due to the selectivity (i.e., PLAP antibody) and specificity (i.e., non-reactive binding platform) of our EV-CATCHER assay, as described in a previous study [66]. These analyses, however, demonstrate that PLAP^+^ EVs can be selectively isolated from maternal plasma and successfully evaluated by NGS analyses of their small-RNA content.

Our small-RNA next generation sequencing analyses of selectively purified PLAP^+^ EVs identified 40 miRNAs that were significantly differentially expressed between women with placenta percreta (cases) and women with placenta previa (controls) pregnancies. When focusing these analyses on miRNAs with the highest read counts (i.e., baseMean > 1000), we identified three upregulated (i.e., miR-151-3p, miR-199a-5p and miR-340) and three down regulated (i.e., miR-122, miR-378 and miR-486) miRNAs. Although we are the first, to our knowledge, to determine that the deregulation of these EV-miRNAs may be associated with percreta pregnancies, recent studies have shown that the deregulated expression of miRNAs (i.e., as EV-miRNAs and circulating miRNAs) is also associated with other adverse pregnancy outcomes [71,72], including preeclampsia and spontaneous pre-term birth [73,74,75,76]. For example, miR-486, which we found to be downregulated in placental EVs of placenta percreta pregnancies, has been found to be upregulated in EVs derived from human placental microvascular endothelial cells, and it is associated with the regulation of proliferation, migration, and invasion of trophoblast cells, thus contributing to poor placentation and the clinical manifestation of preeclampsia [77,78]. Furthermore, in spontaneous pre-term birth, several studies have shown that miR-199a, which we found to be upregulated in placental EVs of placenta percreta pregnancies, plays a critical role in mediating the opposing effects of estrogen and progesterone in uterine contractility during pregnancy, while its downregulation during pregnancy results in spontaneous preterm birth [79,80]. Although the role of these miRNAs (i.e., miR-151, miR-199a, miR-486, miR-122) has not been explicitly studied in the context of placenta percreta, their identification warrants further analyses on their putative role in the context of this pathology. Considering that our EV-CATCHER assay allows for the purification of intact, functional EVs [81], we propose that PLAP^+^ EVs purified from maternal plasma from both percreta and previa pregnancies may be tested to treat cells in vitro in order to identify the putative molecular pathways involved in PAS.

Biologically, the fundamental pathology of PAS involves an overly invasive phenotype of placental cells, which can result in significant potential for maternal morbidity and mortality [82]. When evaluating the literature for a correlation between the differential expression of these top six miRNAs (i.e., miR-486, miR-199a, miR-122, miR-151, miR-378 and miR-340) and cellular invasion, we identified many studies implicating them in the control of cell migration in human cancer cells [82,83,84,85,86,87,88,89]. For example, several studies have demonstrated that miR-486, which we found to be downregulated in placental EVs of placenta percreta pregnancies, is suppressed in different cancer types, including lung, colorectal, and thyroid carcinoma [90,91,92,93], but, when transfected into cancer cells, it leads to a suppression of cell migration, angiogenesis, and invasion [94]. Studies on miR-122, which we found to be downregulated in placental EVs of placenta percreta pregnancies, have shown that, when it is delivered by EVs to both colorectal and breast cancer cells, it significantly reduces their metastatic capacity, and thus its decreased expression in cancer cells has been associated with increased cellular invasion [95]. Moreover, miR-199a, which we found to be upregulated in placental EVs of placenta percreta pregnancies, has been shown to contribute to the progression of malignant tumors, specifically when increased in plasma EVs, and it promotes cellular proliferation and migration of neuroblastoma cancer cells [96]. Interestingly, all six miRNAs have also been associated with the regulation of epithelial-to-mesenchymal transition (EMT), a mechanism common to cancer metastasis [97,98,99,100]. These findings are relevant because we have previously shown that EMT promotes cytotrophoblast to extravillous trophoblast (EVT) differentiation and that the EMT signal is enhanced or prolonged in PAS [11]. Particularly, as we determined that miR-122, miR-378 and miR-486 are downregulated in placental EVs in our PAS cases, it is relevant to note that they have also been reported to exert inhibitory actions on EMT in cancer [86,101,102,103,104,105,106,107,108]. Importantly, miR-151, miR-199a and miR-340, which we found to be upregulated in placental EVs of PAS pregnancies, have been shown to both promote and inhibit the EMT process depending on the cellular context [89,109,110,111,112,113,114,115]. Therefore, the differential expression of EMT-associated placental EV-miRNAs in the context of PAS raises important questions about their putative pathological role during these pregnancies, and thus will warrant further evaluation.

When conducting our miRNA pathway enrichment and REVIGO gene ontology analyses on all 40 differentially miRNAs identified by our NGS analyses, we further found that, globally, these placental EV-miRNAs may regulate genes involved in the promotion of cellular invasion and, potentially, the progression of PAS. Specifically, our miRNet analysis of the top 14 miRNAs upregulated in EVs that we isolated from our PAS show a strong interaction with ZEB1. It is important to note that ZEB1 has been shown to play a pivotal role in enabling proliferation, invasion, and EMT for trophoblast cells during pregnancy [116]. However, the mechanisms by which trophoblast cells achieve these biological effects remain unclear, but we hypothesize that placental EVs transporting these miRNAs may provide a significant stimulus for invasion. Furthermore, we noted that 25 out of the 40 miRNAs found to be differentially expressed in EVs purified from the plasma of our PAS cases belong to the chromosome 19 miRNA cluster, of which eight belong to the miR-500 miRNA family (i.e., miR-545, miR-501, miR-502, miR-519a-1, miR-519a-2, miR-525, miR-518e and miR-512) that have been found to promote invasion [117]. Altogether, our miRNA NGS analyses identified that circulating PLAP+ EVs contain miRNAs that are differentially expressed in placenta percreta pregnancies (i.e., invasive phenotype) in comparison to previa pregnancies, suggesting that circulating placental EVs may contribute to the PAS pathology and, importantly, that they may provide globally quantifiable biomarkers for the detection of this condition.

Although a few studies have evaluated the isolation of placental EVs directly from maternal blood, to our knowledge none have evaluated their isolation and miRNA profiling from the plasma of women diagnosed with placenta percreta. Using molecular assays, we propose that additional analyses will be required to determine how well and how early in pregnancy circulating placental EVs can be captured, as well as how accurately their miRNA cargos may be assessed to detect PAS conditions, in order to evaluate their potential role as clinical biomarkers.

## 4. Materials and Methods

### 4.1. Clinical Specimen Collection

In this study we selected 16 subjects with placenta previa as our controls and 16 subjects with placenta percreta as our cases. Placenta previa pregnancies were selected as our control group for the following reasons: (i) we have previously shown that gene expression profiles in previa and healthy uncomplicated pregnancies are similar [11]; (ii) major risk factors (i.e., age, Cesarean section) for the development of both placenta previa and for PAS are shared; (iii) both previa and PAS are generally delivered pre-term by Cesarean section (CS) without labor for similar medical reasons (bleeding, PPROM, suspected abruption). These factors collectively allowed us to select controls and cases due to (i) similarities in gestational age, (ii) the elimination of labor effects from the molecular analyses [118], and (iii) to account for treatment of delivery indicators (e.g., bleeding) as confounders [119]. This study was approved by the Institutional Review Board (IRB) for Hackensack University Medical Center under protocol Pro00005221, and informed consent was obtained from the subjects prior to sample collection. We enrolled 16 subjects diagnosed with placenta previa pregnancies whose plasma specimens were obtained immediately prior to Cesarean delivery, as well as 16 subjects diagnosed with placenta percreta pregnancies (PAS cases) whose plasma specimens from patients with were obtained prior to Cesarean-hysterectomy; each subject was enrolled through the Center for Abnormal Placentation at Hackensack University Medical Center. All subjects consented to use of blood samples, and maternal samples were matched between cases and controls that had the same approximate gestational age (33–35 weeks). Subject demographics are provided in Appendix A. The plasma samples from each of the placenta previa controls and placenta percreta cases were processed within 1 h of sample collection. Briefly, maternal blood specimens were collected in K-EDTA blood collection tubes, mixed, and stored on ice prior to being centrifuged at 2500× *g* for 10 min. The plasma layer was removed and stored as 500 µL aliquots at −80 °C. Prior to final stratification, placenta previa and percreta cases were identified by antenatal ultrasound and MRI assessment. Prenatal diagnoses were later confirmed by histopathological determination. Sex- and gestational-age specific birthweight centiles were calculated according to Fenton and Kim [120].

### 4.2. Western Blot Analysis

Western blot analyses were conducted to confirm the specificity of the PLAP antibody used to purify PLAP^+^ EVs from maternal plasma. Purified recombinant proteins from both placental specific alkaline phosphatase (PLAP, #NBP2-52266) and alkaline phosphatase, tissue-nonspecific isozyme (ALPL, #2909-AP-010) were purchased from Novus Biologicals (Centennial, CO, USA) and separated on 4–12% polyacrylamide precast mini-PROTEAN TGX gel (Bio-Rad, Hercules, CA, USA, cat#4561086) by sodium dodecyl sulphate polyacrylamide gel electrophoresis (SDS-PAGE). An amount of 5 μL of Precision Plus Protein™ Dual Xtra Prestained Protein Standard (Bio-Rad, #1610377) was loaded and used for gel orientation and determination of molecular weights of separated proteins. Additionally, 10 µg of each purified recombinant protein was loaded, and gels were run at 100 V and 400 mA for 90 min (Power Pac 300, Bio-Rad) in Tris/Glycine/SDS buffer (1X) (Bio-Rad, Hercules, CA, USA, cat#1610732). After the SDS-PAGE run, proteins were transferred to 0.2 µm polyvinylidene fluoride (PVDF) membranes (Bio-Rad, Hercules, CA, USA, cat #1704156) using a semi-dry electro-transfer system (TransBlot Turbo v1.02, Bio-Rad, Hercules, CA, USA) for 30 min at 25 V. Membranes were visualized using the stain-free blot protocol provided on a Chemi-Doc™ MP (Bio-Rad, Hercules, CA, USA) imaging system to evaluate protein transfer, while membranes were blocked using EveryBlot blocking buffer (Bio-Rad Hercules, CA, USA, cat#12010020) for 30 min. Membranes were incubated at 4 °C o/n with TBS-T (TBS (1X), pH 6.8, 0.1% Tween20) diluted anti-mouse primary antibodies (1:1000) targeted against PLAP (Novus Biological, Centennial, CO, USA, cat#NBP2-47993) and ALPL (Novus Biologicals, Centennial, CO, USA, cat#NBP2-22193). Membranes were washed with TBS-T (3 × 5 min) before incubation in anti-mouse IgG horseradish peroxidase conjugated secondary antibodies (Abcam, Cambridge, UK, cat#ab6728: 1:10,000) for 1 h, with gentle agitation at RT. Membranes were washed with TBS-T (3 × 5 min) before proteins were detected using SuperSignal™ West Femto Maximum Sensitivity Substrate (ThermoFisher, Waltham, MA, USA, cat#34095), and protein bands were visualized using ImageLab 4.0 software on the Chemi-Doc MP imaging system.

### 4.3. EV-CATCHER Isolation of PLAP^+^ Small-Extracellular Vesicles

The isolation of placental EVs was performed using the EV-CATCHER isolation protocol described by Mitchell et al., 2021, using PLAP (placental alkaline phosphatase) as the capture antibody [68]. Briefly, equimolar amounts (1:1 ratios) of oligonucleotides (Integrated DNA Technologies, San Diego, CA, USA) 5′-Azide(AAAAACGAUUCGAGAACGUGACUGCCAUGCCAGCUCGUACUAUCGAA) and 3′-Biotin(GAUAGUACGAGCUGGCAUGGCAGUCACGUUCUCGAAUCGUUUU) were annealed (90 °C for 2 min, 90–42 °C for 40 min, 42 °C for 120 min) in 1x RNA annealing buffer (60 mM KCl, 6 mM HEPES (pH 7.5), 0.2 mM MgCl_2_) prior to separation on a 15% non-denaturing polyacrylamide (PAGE) gel (450 volts for 90 min). The double stranded (ds) DNA linker was visualized on a blue light box with SYBR^®^ Gold™ dye (ThermoFisher, Waltham, MA, USA, cat#S11494), excised, centrifugally crushed using a gel breaker tube (IST Engineering, Milpitas, CA, USA, cat#3388-100), and resuspended in 400 mM NaCl and shaken overnight on a thermomixer set to 4 °C and 1100 RPM. The solution was filtered, and the dsDNA linker was purified using the QIAEX^®^ II gel extraction kit (Qiagen, Hilden, Germany, cat#20021) according to manufacturer instructions. The anti-PLAP antibody (1 mg/mL) used for EV pulls (Novus, Centennial, CO, USA, cat #NBP2-47993), was activated using 5 µL of freshly prepared 4 mM DBCO-NHS ester (Lumiprobe, Cockeysville, MD, USA, cat#94720) and incubated for 30 min at room temperature (RT) in the dark, and reactions were stopped by adding 2.5 µL of 1 M Tris-Cl (pH 8.0) at RT for 5 min in the dark. DBCO-activated anti-PLAP was then desalted onto pre-equilibrated Zeba desalting columns (ThermoFisher, Waltham, MA, USA, cat#89882) by incubation for 1 min and centrifugation at 1500× *g* for 2 min. Purified dsDNA linker and DBCO activated anti-PLAP antibody were quantified on a Nanodrop 2000 instrument prior to the preparation of antibody-dsDNA (Ab-dsDNA) stock solutions, where 100 μg of activated antibody was conjugated to 50 μg of purified DNA linker, and incubated o/n at 4 °C on a rotator. The next day, Ab-dsDNA conjugates were bound to streptavidin coated 96-well plates (ThermoFisher, Waltham, MA, USA, cat#15120) by incubating 1 µg of anti-PLAP antibody (linker bound) in 100 µL 1x PBS per well (2 wells were prepared per sample). Plates were then placed on a plate shaker at 300 RPM at 4 °C for 8 h to allow for binding to the plate. Solutions were carefully removed, and wells were washed three times with cold PBS (1x) solution prior to addition of RNase-A (12.5 μg/mL) treated samples (100 μL). Plates were then sealed using microAMP optical adhesive film (Applied Biosystems, Waltham, MA, USA, cat#4311971) and placed on a shaker at 300 RPM at 4 °C, o/n. Samples were carefully removed, wells were washed 3 times with cold PBS (1x), and 100 μL of freshly prepared uracil glycosylase (UNG) enzyme (ThermoFisher, Waltham, MA, USA #EN0362) in PBS (1x) (UNG (1x) buffer (200 mM Tris-Cl (pH 8.0), 10 mM EDTA and 100 mM NaCl), with 1 unit of enzyme) was added to each well. Plates were incubated at 37 °C for 2 h on a shaker at 300 RPM for UNG digestion of the dsDNA linker, and PLAP^+^ EVs were recovered in this solution for downstream miRNA analyses.

### 4.4. Transmission Electron Microscopy

Transmission electron microscopy (TEM) of EVs was performed at the analytical imaging facility at the Albert Einstein College of Medicine, Bronx, NY. Briefly, purified EVs were fixed using 2% Glutaraldehyde in phosphate buffer (Electron Microscopy Services, Hatfield, PA, USA, cat#6536-05) and stored at 4 °C. A total of 300 mesh formvar-coated grids were inverted onto 20 µL of fixed EV suspensions for approximately 2 min and wicked dry. Grids were then inverted onto 40 µL of 2% aqueous uranyl acetate for approximately 1 min and then wicked dry. Samples were imaged on a JEOL JEM-1400+ transmission electron microscope (JEOL Ltd.; Tokyo, Japan) operating at an accelerating voltage of 80 kV. High resolution TIFF images were acquired and saved using an AMT 16 MP digital camera system (Advanced Microscopy Techniques Corp.; Woburn, MA, USA).

### 4.5. ONi Super Resolution Nanoimaging

Purified PLAP^+^ (positive) EVs were processed for imaging on the ONi super resolution Nanoimager using the ONi EV Profiler kit v2.0 according to manufacturer’s instructions (ONi, San Diego, CA, USA). Briefly, the surface of the assay capture chip was prepared by applying 5 µL of S3 buffer to each lane and incubated at room temperature for 10 min. An amount of 30 µL of W1 was then applied to each lane to remove excess S3, after which 10 µL of S4 buffer was slowly pipetted to each lane, ensuring that no bubbles were introduced into lanes. After a 10 min incubation period at room temperature, lanes were again washed by applying 30 µL of W1 buffer to each lane. EV capture was then performed by immediately applying 10 µL of EV-CATCHER purified PLAP^+^ EVs and allowing binding to occur for 15 min. Lanes were then washed using 30 µL of W1 buffer, and captured EVs were fixed by adding 20 µL of F1 to each lane and incubating the chip at room temperature for 10-min. The staining of captured EVs was performed by firstly preparing a three-antibody working solution consisting of CD63-Fluor^®^ 568 (ONi, San Diego, CA, USA. cat#EV-Man-1.0), CD9-Alexa Fluor^®^ 488 (ONi, San Diego, CA, USA. cat#EV-Man-1.0), and PLAP-Alexa Fluor^®^ 647 (Novus Biologicals, Centennial, CO, USA, cat#NBP2-76654AF647) antibodies combined together in W1 buffer so that each antibody is at a dilution of 1:20. The final staining solution was prepared by combining 1 µL of the prepared working solution with 9 µL of N1 buffer for each lane, gently pipetting to mix the solution and applying 10 µL to each lane of the EV profiler chip; it was then allowed to incubate for 50 min at room temperature in the dark. Immediately following antibody incubation, lanes were washed with 30 µL of W1 buffer, followed by a 20 min incubation with 20 µL of F1 buffer for 10 min. A final wash step was performed, and BCubed^TM^ dSTORM imaging buffer was added to each well immediately before EV profiler chips were imaged. Image acquisition on the ONi super resolution Nanoimager was performed in the NimOS Light program with a 640 dichroic split using the following parameters: 640 nm laser set to 20–30% laser power, the 560 nm laser at 35% laser power, and the 473/488 nm laser set to 70% laser power. The number of runs (frames) for all laser lines was set to 1000, and all image analyses were performed using CODI software (ONi, San Diego, CA, USA).

### 4.6. RNA Extractions

PLAP^+^ (positive) EVs isolated from maternal plasma were subjected to total RNA extraction using the miRNeasy Serum/Plasma kit (Qiagen, Hilden, Germany, cat#217184) according to manufacturer’s instructions, with some modifications being added to improve total RNA yield. Briefly, QIAzol was added to 100 µL of PLAP^+^ EVs, vortexed, and incubated at RT for 5 min, after which chloroform was added to each sample. Samples were vortexed again and incubated at RT for 5 min. Samples were centrifuged at 12,000× *g* at 4 °C for 15 min, and the upper aqueous phase of each sample was then carefully removed and transferred into new siliconized tubes, to which 100% ethanol was added. Samples were incubated on ice for 40 min prior to column purification. The clear upper phase was then passed twice through supplied RNeasy minElute columns and washed with RPE and ice cold 80% ethanol. Columns were spun to remove residual ethanol, and total RNA was eluted with 50 µL of RNase-free water. Samples were then speed-vacuumed to 10 µL prior to small-RNA sequencing.

### 4.7. Small-RNA cDNA Library Preparations

Small-RNA sequencing from PLAP^+^ (positive) EVs was performed using the cDNA library preparation protocol described by Loudig et al. (2017), with modifications being added for low input RNA from purified EVs [68,121,122]. For PLAP^+^ EVs purified from previa and percreta plasma, the small-RNA cDNA library preparation was performed using total RNA recovered from EVs purified from 200 µL of plasma (2 wells were prepared for each serum sample (100 µL plasma per well), and isolated EVs were pooled for RNA extraction). Samples were divided into two libraries, each containing 8 placenta percreta and 8 previa samples (a total of 16 samples per library). A total of16 ligations were set up individually by combining 9.5 µL of total RNA, 8.5 µL of the master-mix, and 1 µL of 50 µM adenylated barcoded 3′ adapter (Integrated DNA Technologies, San Diego, CA, USA; custom order). A master-mix was then prepared using a 0.0052 nM calibrator cocktail (40 µL 10x RNA ligase-2 buffer, 120 µL 50% DMSO, and 10 µL calibrator cocktail). Reactions were heated at 90 °C for 1 min, incubated on ice for 2 min, and 1 µL (1/10 diluted) truncated K227Q T4 RNA Ligase 2 (New England Biolabs, Ipswich, MA, USA, cat#M0351L) was added to each reaction, which were then incubated O/N on ice in a cold room. The next day, ligations were heat inactivated at 90 °C for 1 min and individually precipitated by the addition of 1.2 µL of Glycoblue mix (1 µL Glycoblue™ Co-precipitant (15 mg/mL; ThermoFisher, Waltham, MA, USA, cat#AM9516) in 26 µL 5 M NaCl (ThermoFisher, Waltham, MA, USA, cat#AM9579)) before 63 µL of 100% ethanol was added to each tube. Reactions were combined, precipitated on ice for 1 h, and centrifuged for 1 h at 14,000 RPM at 4 °C. The pellet was dried and resuspended in 20 µL nuclease-free water and 20 µL denaturing PAA gel loading solution before being separated on a 15% Urea-PAGE gel. Size marker RNA oligonucleotides (IDT) were used to guide gel excision. The gel piece was crushed using a gel-breaker tube (IST Engineering, Milpitas, CA, USA, cat#3388-100) and incubated in 400 mM NaCl O/N at 4 °C at 1100 RPM on a thermomixer. The next day, the solution was filtered and precipitated in 100% ethanol on ice for 1 h. RNA pellet was obtained by centrifugation at 14,000 RPM for 1 h at 4 °C. The 5′ adapter was added to the resuspended pellet, and T4 RNA Ligase 1 (New England Biolabs, Ipswich, MA, USA, cat#M0204L) was added for 1 h at 37 °C. The ligated product was separated on a 12% Urea-PAGE gel in the presence of 5′ ligated size markers, which acted as a guide for size selection. The gel was spun in a gel breaker tube, after which the crushed gel was resuspended in 300 mM NaCl solution with 1 µL 100 M 3′ PCR primer and incubated O/N on a thermomixer at 1100 RPM at 4 °C. Subsequently, the solution was filtered, precipitated with 100% ethanol, incubated on ice for 1 h, and pelleted by centrifugation at 14,000 RPM for 1 h at 4 °C. The RNA pellet was resuspended in nuclease free water for reverse transcription (3 µL 5x first strand buffer, 1.5 µL of 0.1 M DTT, and 4.2 µL dNTP Mix (2 mM each; ThermoFisher, Waltham, MA, USA, cat#R0241)) with 0.75 µL SuperScript^®^ III Reverse Transcriptase (ThermoFisher, Waltham, MA, USA, cat#18080-093) and incubated at 50 °C for 30 min. Reverse transcription was deactivated at 95 °C for 1 min, followed by addition of 95 µL nuclease-free water. A pilot PCR reaction was set up (10 µL PCR buffer (10X), 10 µL dNTP Mix (2 mM each), 10 µL cDNA, 67 µL nuclease-free water, 0.5 µL 3′ PCR primer, 0.5 µL 5′ PCR primer, and 2 µL Titanium^®^ Taq DNA Polymerase (Clontech Laboratories, Mountainview, CA, USA, cat#639208)). An amount of 12 µL aliquots were withdrawn at cycles 10, 12, 14, 16, 18, 20, and 22 for analysis on a 2.5% agarose gel and the identification of the optimal PCR amplification cycle. Six PCR reactions were then set up, run for the optimal number of amplification cycles, and a portion (10 µL) was evaluated on a 2.5% agarose gel. The remaining solution was combined, precipitated, digested with PmeI for removal of size markers, and separated on a 2.5% gel. The 100 nt PCR library product was excised, purified with QIAquick Gel Extraction Kit (Qiagen, #28704), and quantified using the Qubit^®^ dsDNA HS Assay Kit (ThermoFisher, Waltham, MA, USA, #Q32854). Additionally, cDNA libraries were then sequenced (single-read 50 cycles) on a HiSeq 2500 Sequencing System (Illumina, San Diego, CA, USA, cat#SY-401-2501), after which FASTQ files containing raw sequencing data were processed for adapter trimming and small-RNA alignment to the hg-19 genome. Read counts were normalized to total counts and subjected to statistical analyses (see below).

### 4.8. In Silico miRNA Enrichment and Gene Ontology Analyses

MiRNA enrichment analysis (miRNet) and REVIGO gene ontology analyses were performed separately for the top 25 upregulated and the top 15 downregulated miRNAs from the list of top 40 differentially expressed miRNAs obtained from miRNA sequencing. Based on the list of the top 40 differentially expressed miRNAs between percreta cases and previa controls, miRNAs were further stratified by disregarding any miRNAs where the baseMean (readcount) fell below 100, identifying 23 miRNAs. The resulting list of 23 differentially expressed miRNAs was then used for all subsequent enrichment analyses and gene ontology pathway predictions. Based on the fold-change statistics generated, the miRNAs were classified as either upregulated or downregulated, and miRbase ID were obtained from the miRbase website (https://mirbase.org. Accessed: 6 December 2022). Additionally, for all mature miRNAs, GO terms from miRBase were also obtained. For these analyses, the top 14 upregulated and top 9 downregulated miRNAs were assessed separately in miRNet (https://www.mirnet.ca/upload/MirUploadView.xhtml, accessed on 22 February 2024), where each of the upregulated and downregulated miRNAs were imputed using their miRBase IDs while selecting miTRarBase v8.0 as a target. The interaction networks for both the upregulated and downregulated miRNAs were filtered using a degree filter of 2.0 for all network nodes, and the minimum network selection was then selected. Two separate comprehensive lists (one for upregulated and one for downregulated miRNAs) of GO Terms were obtained from the miRbase. For all mature differentially expressed miRNAs, overlapping GO terms were consolidated, and each list was separately imputed into REVIGO (http://revigo.irb.hr/. Accessed: 12 December 2022) to visualize the predicted gene ontology pathways.

### 4.9. Data Analysis

Raw FASTQ data files obtained on an Illumina HiSeq2500 sequencer were processed using the RNAworld server from the Tuschl Laboratory at the Rockefeller University, including adapter trimming and read alignments and annotation. MiRNA counts were exported to spreadsheets for data analysis. Statistical analyses of miRNA counts were performed using dedicated Bioconductor packages in the R platform, as detailed below. Heat maps were generated from transformed counts using the ‘NMF’ package (a heatmap function). Differential expression was assessed using ‘DESeq2’ and ‘edgeR’. Differential expression models included a batch variable (library) to reduce batch biases. To maximize the discrimination ability of each miRNA, we computed a score for each sample (‘miRNA score’, [123]) that was assembled by summing the standardized levels (*z*-values) of all significantly upregulated miRNAs and the negative of the *z*-values of all significantly downregulated miRNA.

## Figures and Tables

**Figure 1 ijms-25-05102-f001:**
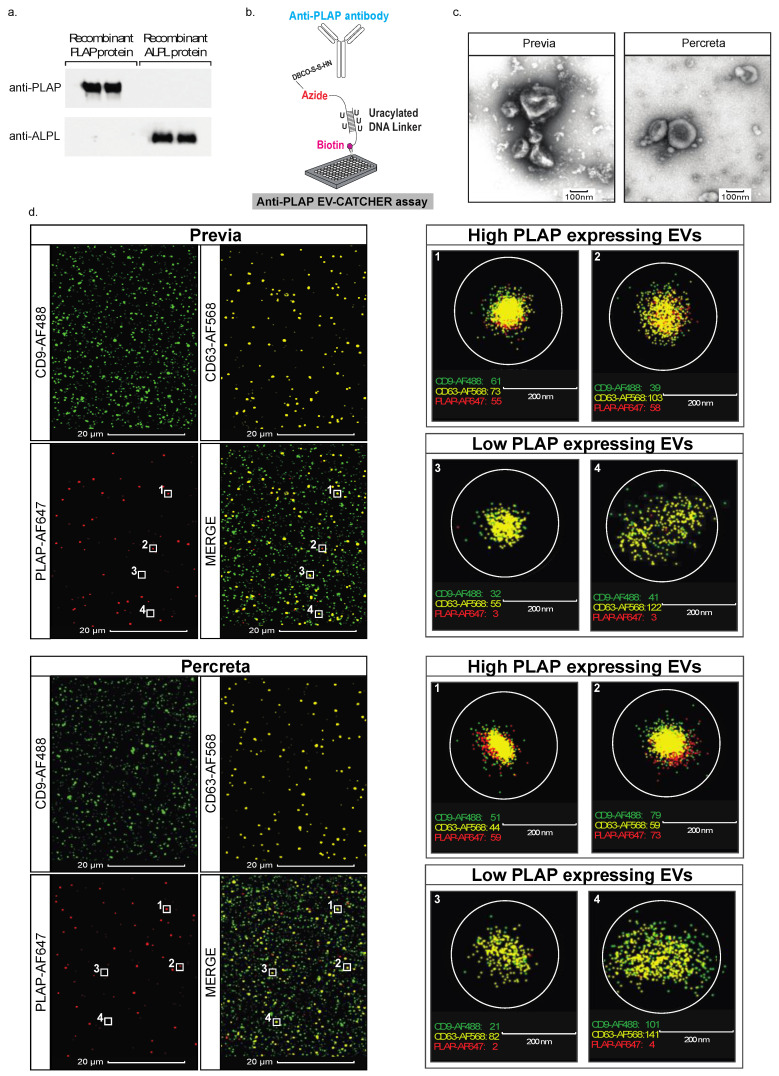
Validation of EV-CATCHER customized antibodies and characterization of placental EVs. (**a**). Western blots displaying the specificity of the monoclonal placental alkaline phosphatase antibody against the PLAP recombinant protein (Novus, Centennial, CO, USA, #NBP2-52266) used for EV-CATCHER customization versus the alkaline phosphatase, tissue-nonspecific isozyme antibody against the ALPL recombinant protein (Novus, Centennial, CO, USA, #2909-AP-010). (**b**). EV-CATCHER assay customized with an anti-PLAP antibody conjugated to a uracilated DNA linker that is connected by a biotin to the bottom of a streptavidin coated well. Using uracil DNA glycosylase (UNG), the immobilized EV-anti-PLAP duplex is released by enzymatic digestion of the double-stranded uracilated DNA linker. (**c**). Representative Transmission electron microscopy (TEM) images of EVs isolated from placenta previa and placenta percreta maternal plasma, scale bar 100 µm. (**d**). Representative ONi super resolution images of EVs isolated from placenta previa and placenta percreta maternal plasma displaying both high PLAP^+^ expressing EVs and low PLAP^+^ expressing EVs, scale bar of 20 µm for large field-of-view images and 200 nm for individual representative EVs.

**Figure 2 ijms-25-05102-f002:**
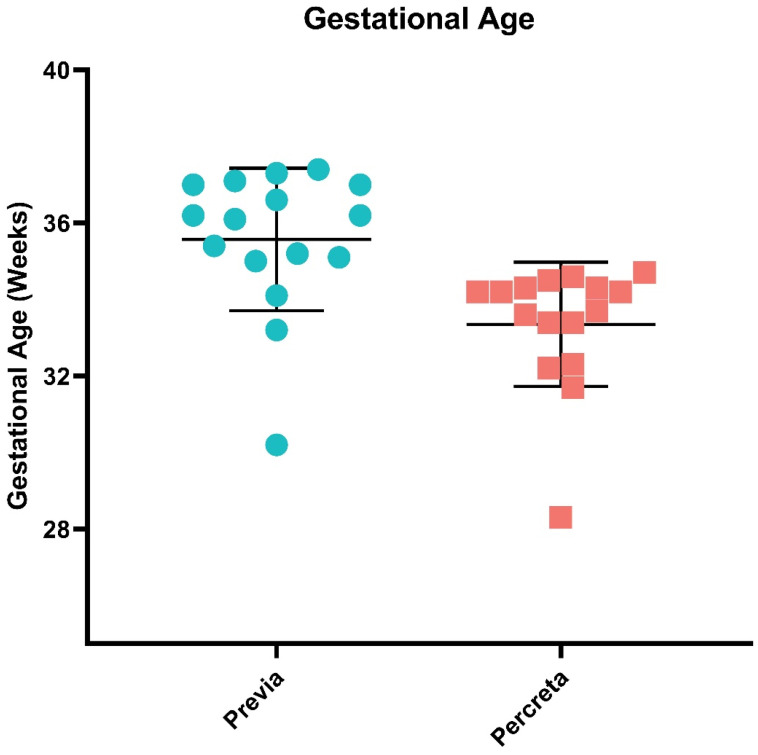
Mean gestational age of women with placenta previa and percreta pregnancies selected for this study. Prenatal diagnoses were confirmed by histopathological determination after delivery of the placenta following elective Cesarean section (placenta previa) or after Cesarean hysterectomy (placenta percreta).

**Figure 3 ijms-25-05102-f003:**
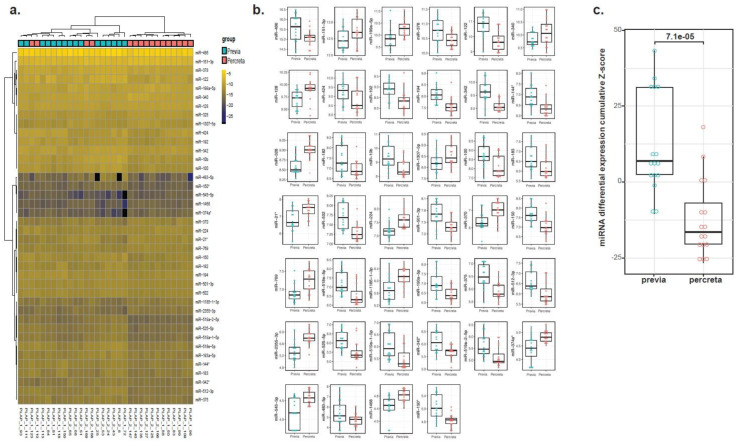
Small-RNA sequencing of placental EVs purified from maternal plasma using the PLAP customized EV-CATCHER assay. (**a**). Heatmap representation displaying miRNA expression differences between total RNA extracted from PLAP^+^ EVs from previa (lanes 1–2, 5–12 and 15–20) and percreta (lanes 3–4, 13–14 and 21–32). (**b**). Individual box plot analyses of the 40 differentially expressed miRNAs identified from total RNA extracted from PLAP^+^ EVs purified from maternal plasma with the EV-CATCHER assay. (**c**). Box plot representation of the integrative miRNA signature (40 miRNAs) between total RNA extracted from PLAP^+^ EVs purified EV-CATCHER from subjects with placenta previa (*n* = 16) and placenta percreta (*n* = 16).

**Figure 4 ijms-25-05102-f004:**
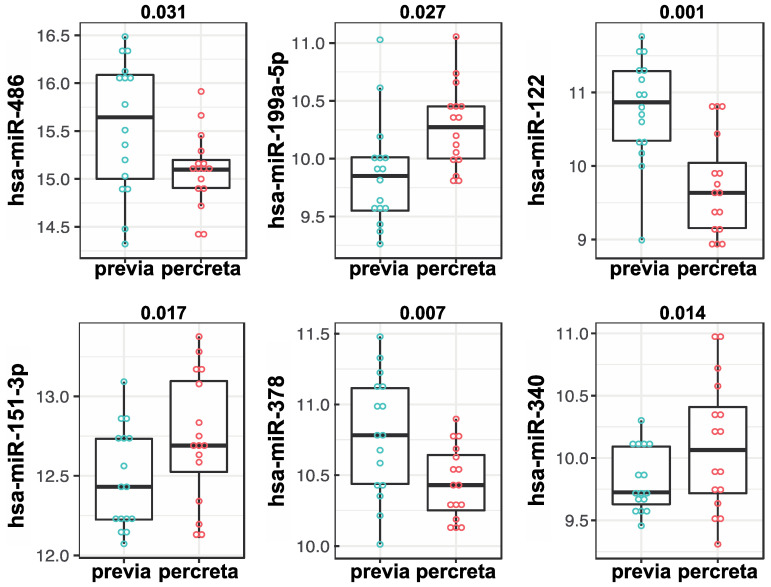
Identification of top six differentially expressed miRNAs with the highest reads in placenta percreta. Individual box plot analyses of the six top differentially expressed miRNAs identified from total RNA extracted from PLAP^+^ EVs purified from maternal plasma with the EV-CATCHER assay (hsa-miR-486, hsa-miR-151-3p, hsa-miR-199a-5p, hsa-miR-378, hsa-miR-122, hsa-miR-340).

**Figure 5 ijms-25-05102-f005:**
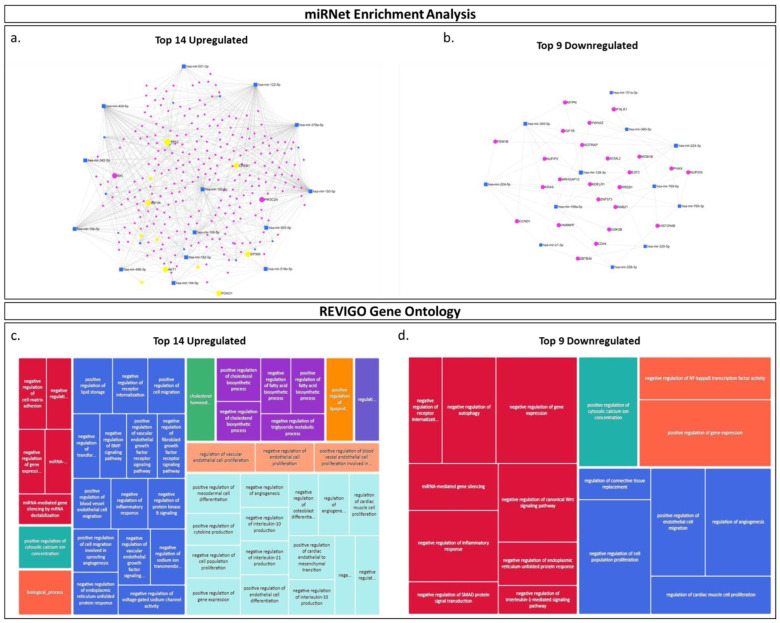
The miRNA enrichment and gene ontology analyses of top upregulated and downregulated miRNAs in placenta percreta. The miRNet network analyses displaying the predicted microRNA-gene interactions for the (**a**) top 14 upregulated and (**b**) the top nine downregulated EV-miRNAs identified by NGS. Each analysis in miRNet utilized miTRarBase v8.0 as the selected gene target database, and filtering was performed using a degree filter of 2.0 for all network nodes and the minimum network selection. REVIGO gene ontology tree maps displaying the predicted biological processes in which the (**c**) top 14 upregulated and (**d**) top nine downregulated EV-miRNAs are involved. Comprehensive lists of GO terms were obtained from miRBase for all mature differentially expressed miRNAs, and overlapping GO terms were consolidated and imputed into REVIGO.

## Data Availability

Data is contained within the article or Appendix A.

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
