# Peer review of "Customizing EV-CATCHER to Purify Placental Extracellular Vesicles from Maternal Plasma to Detect Placental Pathologies"

_ijms, 2024, doi:10.3390/ijms25105102_

Round 1

Reviewer 1 Report (Previous Reviewer 2)

Comments and Suggestions for Authors

In this manuscript titled “Customizing EV-CATCHER to purify placental specific extracellular vesicles from maternal plasma to detect placental pathologies, the authors purify placental EVs containing PLAP from maternal blood and analyzed the miRNAs present in the EVs.

The manuscript presents some interesting findings, but the PLAP EV-Catcher experiment lacks appropriate negative controls. It is important for the authors to prove that placental EVs with PLAP present could be purified from maternal blood.

Comments:

1.     Figure 1: The number of PLAP red dots is considerably less than the number of CD9 and CD63; PLAP-negative and CD9-positive vesicles would also have been acquired by the EV-Catcher experimental system; the same is true for PLAP-negative and CD63-positive vesicles. Did the authors really acquire PLAP-specific EVs? The reviewer believes that the authors' EV isolation method contains a significant amount of PLAP-negative EVs based on the results in Figure 1c. The authors should provide the percentage of overlap of each color. The authors should then discuss the results of experiments such as small RNA sequencing (Figure 3-5) using EVs containing PLAP negatives. The reviewers will be asked to discuss the issues, including a significant amount of PLAP negative EVs, from a multiple perspective, including improvements to the methodology.

2.     The authors successfully obtained EVs from plasma samples from patients with placental abruption (percreta) (cases) and placenta previa without adherent placenta (controls). However, this is not enough to demonstrate successful isolation of placental-specific EVs. The authors should either change the title or provide the control data necessary for successful isolation of placenta specific EVs.

Author Response

REVIEWER 1

In this manuscript titled “Customizing EV-CATCHER to purify placental specific extracellular vesicles from maternal plasma to detect placental pathologies”, the authors purify placental EVs containing PLAP from maternal blood and analyzed the miRNAs present in the EVs.

The manuscript presents some interesting findings, but the PLAP EV-Catcher experiment lacks appropriate negative controls. It is important for the authors to prove that placental EVs with PLAP present could be purified from maternal blood.

Comments:

  1. Figure 1: The number of PLAP red dots is considerably less than the number of CD9 and CD63; PLAP-negative and CD9-positive vesicles would also have been acquired by the EV-Catcher experimental system; the same is true for PLAP-negative and CD63-positive vesicles. Did the authors really acquire PLAP-specific EVs? The reviewer believes that the authors' EV isolation method contains a significant amount of PLAP-negative EVs based on the results in Figure 1c. The authors should provide the percentage of overlap of each color. The authors should then discuss the results of experiments such as small RNA sequencing (Figure 3-5) using EVs containing PLAP negatives. The reviewers will be asked to discuss the issues, including a significant amount of PLAP negative EVs, from a multiple perspective, including improvements to the methodology.

Response:

We wish to thank Reviewer#1 for the thorough review of our manuscript and for the insightful comments, which we have addressed and that have ultimately helped to significantly improve our manuscript. While we agree that in our initial submission we did not accurately represent the validations of our PLAP customized EV-CATCHER assay, which visually suggested that we isolated a high number of non-specific EVs (i.e., when comparing the number of CD9 green clusters to those of PLAP red clusters) we have provided explanations in the text of our Results and Discussion sections and are added matching details in Figure 1 to clarify this observation. Specifically, we have included representative ONi images of individual EVs (i.e., focused on individually immobilized EVs), which now display: 1- high PLAP expressing EVs (i.e., EVs with >50 positive PLAP copies on their surface) that can also be observed on the large field-of-view panels (i.e., similarly to CD9 green clusters), but also importantly 2- low PLAP expressing EVs (i.e., <5 positive PLAP copies on the surface of immobilized EVs) that cannot be observed from the large field-of-view panels that were selectively isolated from the plasma samples of both placenta previa and placenta percreta pregnancies. We provided matching numbers (i.e., 1, 2 for high PLAP and 3, 4 for low PLAP expressing EVs) in both the large field-of-view and the focused EV images so readers may visually match the position of high and low expressing EVs between both point of views. Although focused images of low expressing EVs display the presence of single red dots (i.e., fluorescent signal for detection of PLAP), it clearly shows that when match to the large field-of-view, these single dots cannot be observed and thus convey the wrong impression that we have a large number of PLAP negative EVs. We hope that Reviewer#1 will find these images sufficient for the readers. However, we did observe very few EVs or possibly membrane fragment of EVs, immobilized on the ONi platform, which did not display individual dots. Thus, to be transparent with Reviewer#1 and with the readers, we provided detailed explanation for these overall rare events on the ONi super resolution nanoimages. We detailed three explanations in the Discussion section of the manuscript suggesting the possibility that: i) several PLAP epitopes located on surface of immobilized EVs may be blocked from physical access by the fluorescently label PLAP antibody due to the permanent attachment of the PLAP antibody released from the EV-CATCHER platform after their isolation from plasma. We have provided a schematic of the EV-CATCHER assay in Figure 1b with details of the biochemical release of the captured EVs in the Figure’s legend, which will help readers understand how the EV-CATCHER assay works and how the antibody is released from the platform with its captured EV; ii)  PLAP proteins present on these EVs may be localized to the opposite side of the immobilized EV, which faces the ONi platform; and iii) some EVs may have been non-specifically captured, which is very unlikely as we described in very precise details and experiments how the EV-CATCHER is much more selective than 10 other commercially and laboratory-based EV purification assays in our Mitchell MI et al 2021 manuscript in the Journal of Extracellular Vesicles (JEV). Indeed, we were able to demonstrate, using qPCR experiments that the EV-CATCHER assay had extremely low non-specific EV capture ability when compared to magnetic beads. Altogether our experiments and the foundational experiments that we detailed in Mitchell MI et al, 2021 demonstrate that the selectivity of our customizable EV-CATCHER assay (i.e., selection of a monoclonal antibody targeting a single epitope) is uniquely dependent on the antibody’s specificity, which we confirmed in our Western blot analysis in Figure 1a. In order to address the comments provided by Reviewer#1 in the manuscript we have provided the following material and made the following changes:

Figure 1:

  • This figure has been updated to include a new panel (Fig. 1b) where we display a visual representation of our PLAP customized EV-CATCHER assay. We have specifically included this panel in the manuscript in order to display to readers that following our uracyl DNA glycosylase digest of the DNA linker connecting the antibody to the well of our platform, the PLAP antibody remains attached to the EVs and thus has the physical ability to block access to the PLAP epitopes, especially for EVs with very few PLAP proteins on their surface, during our ONi super resolution nanoimaging analyses.
  • We have also modified Figure 1d to include both high PLAP expressing individually immobilized EVs (i.e., as 1 and 2 in left and right panels) and low PLAP expressing individually immobilized EVs (i.e., as 3 and 4 in left and right panels) isolated from the plasma of placenta previa and placenta percreta pregnancies. This modification allows readers to observe that even though it appears that large clusters of the abundantly expressed EV-specific CD9 tetraspanins (i.e., large field-of-view with green fluorescence) visually identify many EVs, which PLAP protein clusters partially match in the large field-of-view, many of the PLAP positive EVs contain low number of PLAP proteins that cannot be observed in the large field-of-view. Hence, we matched individual images of immobilized EVs on the right of Figure 1d to the exact areas of the large field-of-view ONi platform with numbers 1 and 2 for high PLAP expressing EVs, and 3 and 4 for low PLAP expressing EVs.

Figure 1 Legend:  

  • Lines 420 to 432. We modified the legend of Figure 1 to accommodate the inclusion of a graphical depiction of the EV-CATCHER assay and additional panels for our ONi super resolution nanoimaging as follows:

- “b. EV-CATCHER assay customized with an anti-PLAP antibody conjugated to a uracilated DNA linker that is connected by a biotin to the bottom of a streptavidin coated well. Using uracil DNA glycosylase (UNG) the immobilized EV-anti-PLAP duplex is released by enzymatic digestion of the double-stranded uracilated DNA linker.“  

- “d. Representative ONi super resolution images of EVs isolated from placenta previa and placenta percreta maternal plasma displaying both high PLAP+ expressing EVs and low PLAP+ expressing EVs, scale bar of 20µm for large field-of-view images and 200 nm for individual representative EVs.“ 

Abstract:

  • Line 27. We have removed the term selectively from the sentence “Thus, to purify placental EVs…”
  • Line 35. We modified the sentence from “placental specific EVs…” to “…PLAP+ EVs…”

Results:

  • Lines 394 to 417. In order to clarify to readers the help with understanding of the principle of the ONi super-resolution nanoimaging platform we added the following paragraph “…Moreover, our ONi super resolution nanoimaging analyses, which evaluated the localization of CD9, CD63 and PLAP on immobilized EVs isolated from the plasma of women with previa and percreta pregnancies, using our PLAP EV-CATCHER assay, demonstrated equal distribution levels of these three surface markers between the two different plasma samples (Fig.1d large field-of-view images). We observed that the EV-specific CD9 protein (Fig. 1d, green fluorescent signal on left panels for previa and percreta plasma samples) was the most abundant tetraspanin detected on the surface of immobilized EVs but appeared similarly expressed on the surface of EVs purified from our two plasma samples. We observed that CD63 was also detectable on most of the immobilized EVs, but it appeared less abundantly expressed on the isolated EVs (Fig. 1d, yellow fluorescent signal on left panels for previa and percreta plasma samples). Finally, we observed the presence of PLAP on the surface of immobilized EVs, but noted that protein clusters in comparison to CD9 and CD63, were less abundant for PLAP proteins (Fig. 1d, red fluorescent signal on left panels for previa and percreta plasma samples). We display that although not observable on the large-scale view (Fig. 1d, left panels for previa and percreta plasma samples), when focusing our observation on single EVs, we confirm the presence of high PLAP expressing EVs but also identified low-PLAP expressing EVs in both previa and percreta plasma samples (Fig. 1d, right panels for previa and percreta plasma samples).”

Discussion:

  • Lines 531 to 552. We added the following paragraph discussing the results obtained using the ONi super-resolution imaging platform and the relevance of our results as they pertain to the isolation of PLAP+ EVs “…Prior to conducting our miRNA analyses, we validated the selectivity of our PLAP antibody customized EV-CATCHER assay for the purification of EVs. We used TEM to confirm size and morphology and ONi super-resolution nanoimaging to confirm the distribution of the EV-specific tetraspanins CD9 and CD63, which appeared to be similarly detected between EVs isolated from both the plasma of previa and percreta pregnancies. Using ONi, we also confirmed the presence of the PLAP proteins on the sur-face of these EVs. We observed that PLAP generally appeared in lower copies on individual EVs than CD9, which revealed to be in large clusters on these EVs. In the large field-of-view images, we easily observed high PLAP expressing EVs (i.e., > 50 PLAP copies), while low PLAP expressing EVs (i.e., < 5 PLAP copies) were not observable. However, when zooming in on individually immobilized EVs, we noted that only a very few EVs did not display any PLAP proteins on their surface. This could be explained by a lack of access to the PLAP epitope on the surface of these EV, which may be due to a blockage of the epitope by the PLAP purifying antibody (i.e., released with the EVs after EV-CATCHER purification, or a lack of physical access to PLAP proteins localized on the side of the EV that was immobilized onto the ONi platform. It is possible that some EVs could have been non-specifically purified, but very unlikely, due to the selectivity (i.e., PLAP antibody) and specificity (i.e., non-reactive binding platform) of our EV-CATCHER assay, as described in a previous study [68]. These analyses however demonstrate that PLAP+ EVs can be selectively isolated from maternal plasma and successfully evaluated by next-generation analyses for their small-RNA content.”
  1. The authors successfully obtained EVs from plasma samples from patients with placental abruption (percreta) (cases) and placenta previa without adherent placenta (controls). However, this is not enough to demonstrate successful isolation of placental-specific EVs. The authors should either change the title or provide the control data necessary for successful isolation of placenta specific EVs.

Response:

Thank you for providing this comment and we wish to direct Reviewer#1 to the detailed responses that we provided in comment 1 (above). In the introduction of our manuscript, we have provided detailed information and cited peer reviewed references on the placental alkaline phosphatase (PLAP) protein. This protein belongs to the transmembrane alkaline phosphatase family but it is unique to the placenta because its mRNA contains a deletion in its 5’ region, which leads to a loss of the last 24 amino-acids in its N-terminal region of the protein, which differentiates it from all other alkaline phosphatases. PLAP is uniquely expressed during pregnancy by placental cells, and it provides unique epitopes that allow monoclonal antibodies raised against it to uniquely detect it, as exemplified in Figure 1a. Antibody manufacturers provide a datasheet that thoroughly detail the specificity of the PLAP antibody. We realize that this was not clearly described in our initial manuscript, and thus have provided additional details in the Introduction, Results, Discussion, and reference section of our manuscript to provide clarity to the readers as follows:

Manuscript Title:

In order to be responsive to the comment provided by Reviewer#1, we have slightly modified the title as follows:

  • We modified the title from “Customizing EV-CATCHER to purify placenta-specific extracellular vesicles from maternal plasma to detect placental pathologies.” To “Customizing EV-CATCHER to purify placental extracellular vesicles from maternal plasma to detect placental pathologies.”

Introduction:

We have reviewed our introduction and further clarified details about the PLAP protein, which is a fetal protein expressed by placental cells during pregnancy. We have verified our references and provided the following text:

  • Lines 119 to 128.” One of the defining characteristics of placental EVs is the presence of placental alkaline phosphatase (PLAP), a transmembrane fetal protein uniquely produced by the placenta [59-61]. Although alkaline phosphatase proteins can be found in all human tissues, as PLAP lacks the last 24 amino acids of its N-terminal region it makes it unique and specific to the placenta, while providing distinct epitopes for its targeted antibody capture. It has been shown that this N-terminal modification increases substrate specificity as well as stability to heat and resistance to chemical inactivation. To date, the main functions of PLAP that have been described include assistance in the transfer of immunoglobulin G (IgG) from the mother to the fetus and the stimulation of fibroblast DNA synthesis and proliferation [62]. As PLAP is known to be a transmembrane protein found in abundance on placental EVs, this unique surface marker has become a target for the purification of placental EVs [63].”

Figure 1:

Our Western blot analyses not only validated the specific recognition of purified recombinant PLAP protein, but also indicates that it does not cross-react with one close family member the recombinant ALPL protein. Details on both PLAP and ALPL are provided in the legend as follows:

Figure 1- Legend:

  • Lines 421 to 424: “ Western blots displaying the specificity of the monoclonal placental alkaline phosphatase antibody against the PLAP recombinant protein (Novus, #NBP2-52266) used for EV-CATCHER customization versus the alkaline phosphatase, tissue-nonspecific isozyme antibody against the ALPL recombinant protein (Novus, #2909-AP-010).”

Results:

We have enhanced our description of the Western blot results in the text of our Results section, as follows to help the readers understand the unique expression of PLAP to placenta and that the antibody we selected does not cross-react with a close family member of surface membrane phosphatase that is expressed in adult humans.

  • Lines 380 to 386: In order to validate our PLAP antibody customized EV-CATCHER assay for the isolation of placental EVs from maternal blood, we first tested our monoclonal PLAP antibody for its ability to recognize the recombinant placental alkaline phosphatase (PLAP) when compared to a recombinant alkaline phosphatase, tissue-nonspecific isozyme (ALPL) protein by Western blot analyses. We demonstrate that our PLAP antibody reacted strongly with the recombinant PLAP protein with no cross-reactivity against the recombinant ALPL protein (Fig.1a).

Discussion:

We have provided information about the PLAP protein at the beginning of the Discussion as follows:

  • Lines 513 to 517: In this pilot study we isolated, analyzed, and profiled the miRNA cargos of placental (PLAP+) EVs isolated from maternal plasma of women with placenta previa (control) and placenta percreta (case) pregnancies, using an antibody targeting the transmembrane placental alkaline phosphatase (PLAP) proteins, which is uniquely produced by placental cells [61].

References:

  1. Sussman HH, Bowman M, Lewis JL Jr. Placental alkaline phosphatase in maternal serum during normal and abnormal pregnancy. Nature. 1968 Apr 27;218(5139):359-60. doi: 10.1038/218359a0. PMID: 5649676.

Reviewer 2 Report (New Reviewer)

Comments and Suggestions for Authors

In this paper, the authors present the a customised EV-CATCHER to purify placental specific extra-cellular vesicles from maternal plasma to detect placental pathologies, such as Abnormally Invasive Placentation. The subject is of great interest in nowadays medicine and can lead to a quicker and better diagnosis of this life-threatening condition, thus saving the life’s of millions of women worldwide.

The introduction focuses on the general knowledge on the subject and provides enough and well-sourced information for a better understanding of the topic.

The authors conducted a proper collection of the data. The information presented is up to date, suitable, and substantial, with cited references from mostly recent publications.

The methods and materials used are well-detailed, complex and reliable, with a study design appropriate to test the hypothesis.

The conclusions are coherent, comprehensive and sustain the findings, that can be seen as a breakthrough in the subject of diagnosing placental abnormalities because of the gap in knowledge identified by demonstrating that the anti-PLAP customised EV-CATHETER assay has the potential to identify biomarkers for non-invasive detection of Placenta Accreta Spectrum. 

The figures and tables presented are easy to interpret and understand, with properly shown data.

Good English level.

I recommend it for publication.

Author Response

REVIEWER 2

In this paper, the authors present the a customized EV-CATCHER to purify placental specific extra-cellular vesicles from maternal plasma to detect placental pathologies, such as Abnormally Invasive Placentation. The subject is of great interest in nowadays medicine and can lead to a quicker and better diagnosis of this life-threatening condition, thus saving the life’s of millions of women worldwide.

The introduction focuses on the general knowledge on the subject and provides enough and well-sourced information for a better understanding of the topic.

The authors conducted a proper collection of the data. The information presented is up to date, suitable, and substantial, with cited references from mostly recent publications.

The methods and materials used are well-detailed, complex and reliable, with a study design appropriate to test the hypothesis.

The conclusions are coherent, comprehensive and sustain the findings, that can be seen as a breakthrough in the subject of diagnosing placental abnormalities because of the gap in knowledge identified by demonstrating that the anti-PLAP customised EV-CATHETER assay has the potential to identify biomarkers for non-invasive detection of Placenta Accreta Spectrum. 

The figures and tables presented are easy to interpret and understand, with properly shown data.

Good English level.

I recommend it for publication.

Response:

We wish to sincerely thank Reviewer#2 for the thorough and kind review of our manuscript. We also wish to indicate that based on comments received from Reviewer#1 and #3, we added additional details that will enhance understanding of the manuscript to readers who may not be versed in the field of placental biology.

Reviewer 3 Report (New Reviewer)

Comments and Suggestions for Authors

Thank you for the opportunity to review the manuscript entitled ‘Customizing EV-CATCHER to purify placental specific extra-cellular vesicles from maternal plasma to detect placental pa-3 thologies’. The authors developed an ultra-sensitive antibody-based EV purification assay, termed EV-CATCHER, and they evaluated the microRNA profiles of placental-specific extracellular vesicles (EVs) circulating in maternal blood in two abnormal placental conditions; placenta previa and placenta percreta, where placenta previa serves as controls. However, the controls are a bit complicated, since both cases and controls are pregnancy pathologies, with similar predestined risk factor profile, but the results in the MS are convincing, significant and has importance. There are some minor concerns, otherwise the manuscript is well written.

I have just some minor remarks:

Line 18 in the Abstract section: EV is not explained.

Line 121: ‘are’ is duplicated

Between lines 124 and 127: reference with nr 62 is cited three times. I think one citation is enough in the end of the section.

Line 167. How do the authors define controls? Placenta previa is an advanced complication and is often concomitant with vaginal bleeding. Did the authors collect placentae after caesarean section both for cases and controls? It is omitted in the Materials and methods section.

Discussion section: The authors did not discuss all the underexpressed and overexpressed miRNA types. Generally, the Discussion section is shorter then the Introduction and it should be extended with the role and function of other types of miRNAs that are not explained.

Author Response

REVIEWER 3

Thank you for the opportunity to review the manuscript entitled ‘Customizing EV-CATCHER to purify placental specific extra-cellular vesicles from maternal plasma to detect placental pathologies’. The authors developed an ultra-sensitive antibody-based EV purification assay, termed EV-CATCHER, and they evaluated the microRNA profiles of placental-specific extracellular vesicles (EVs) circulating in maternal blood in two abnormal placental conditions; placenta previa and placenta percreta, where placenta previa serves as controls. However, the controls are a bit complicated, since both cases and controls are pregnancy pathologies, with similar predestined risk factor profile, but the results in the MS are convincing, significant and has importance. There are some minor concerns, otherwise the manuscript is well written.

I have just some minor remarks:

Comment:

Line 18 in the Abstract section: EV is not explained.

Response:

We apologize for this omission and have corrected this in the abstract by modifying the sentence “… evaluated the potential of circulating placental EVs and their miRNA content…” to “…evaluated the potential of circulating placental extracellular vesicles (EVs) and their miRNA content...”

Comment:

Line 121: ‘are’ is duplicated

Response:

We have removed the duplicate “are” from this sentence

Comment:

Between lines 124 and 127: reference with nr 62 is cited three times. I think one citation is enough in the end of the section.

Response:

Lines 121 to 128. We apologize for the repetition in the in-text citing of reference #62 and have removed the additional in-text citations and have only cited this reference once at the end of the paragraph.

Comment:

Line 167. How do the authors define controls? Placenta previa is an advanced complication and is often concomitant with vaginal bleeding. Did the authors collect placentae after caesarean section both for cases and controls? It is omitted in the Materials and methods section.

Response:

In selection of controls there were three major research design considerations: avoiding the confounding effects of gestational age, mode of delivery (laboring with vaginal or CS delivery versus non-laboring CS), and, insofar as possible, anatomic similarity. Histopathologically confirmed PAS cases at our institution are all delivered by Cesarean-hysterectomy, with a target gestational age of 34-36 weeks. Many deliver earlier, however, due to conditions such as bleeding or pain consistent with abruption. PAS cases in this series comprised 62% complete previa. Placenta previa pregnancies are also delivered pre-term by Cesarean, for reasons similar to PAS. Therefore, using placenta previa pregnancies for controls permits greater rigor in the molecular analyses by anticipating comparable gestational age effects, and eliminating the effects of labor. Finally, by using previa controls, placental location is matched for the majority of PAS cases. In our analyses we are seeking to capture EV-miRNA signatures that are differentially associated with the invasive and proliferative phenotype of PAS, and not confounded by gestational age, labor effects and, insofar as possible, differing location of uterine implantation.

These issues have been discussed in a previous publication and we made sure to highlight this in our Discussion so readers may understand our experimental design [ #11, PMID: 29438480]. We have additionally modified the manuscript to address this.

We provide clarifications on the selection of the controls in the Introduction, Materials and Methods, and Discussion sections as follows:

Introduction:

  • Lines 141 to 146: “We specifically selected placenta previa pregnancies as our controls to avoid the confounding effects of gestational age, mode of delivery (i.e., non-laboring Cesarean vs. laboring with vaginal or Cesarean delivery), and uterine localization of the placenta in order to provide greater rigor in the identification of EV-miRNA signatures uniquely associated with PAS [11].”

Materials and Methods:

  • Lines 150 to 160. We have added the following paragraph “Specifically, in this study we selected 16 subjects with placenta previa as controls and 16 subjects with placenta percreta as cases. Placenta previa was selected as our control group for several reasons, i) we have previously shown that gene expression profiles in previa and healthy uncomplicated pregnancies are similar [11]; ii) major risk factors for the development of both placenta previa and for PAS are shared; iii) both previa and PAS are generally delivered pre-term by Cesarean section (CS) without labor for simi-lar medical reasons (bleeding, PPROM, suspected abruption). Collectively, these factors allowed us to select of controls and cases with i) similarities in gestational age, ii) elimination of labor effects from the molecular analyses [69] and iii) allows for the treatment of delivery indicators (e.g., bleeding) as confounders [70].”
  • Lines 162 to 166. In order to highlight the fact that gestational age played a role in our decision to utilize placenta previa as our control group we include the following sentence “…We enrolled 16 subjects diagnosed with placenta placenta previa, whose plasma specimens were obtained immediately prior to Cesarean delivery, and 16 subjects diagnosed placenta percreta (PAS cases), whose plasma specimens from patients with were obtained prior to Cesarean-hysterectomy…”

Discussion:

  • Lines 522 to 528. We added the following paragraph “…The decision to utilize patients diagnosed with placenta previa as controls for this study was guided by the fact that the presence of placenta previa alone, regardless of uterine scarring, increases PAS risk by 100-fold, as previously described [77,78]. Indeed, im-plantation of the placenta over the cervical ostium (i.e., placenta previa) combined with Cesarean scarring increases the underlying risk of PAS by >10% and each subsequent Cesarean section more than doubles the risk for the development of PAS. [77,79].”

Comment:

Discussion section: The authors did not discuss all the underexpressed and overexpressed miRNA types. Generally, the Discussion section is shorter then the Introduction and it should be extended with the role and function of other types of miRNAs that are not explained.

Response:

We agree with Reviewer#3’s comment and have endeavored to provide details on the most significant miRNAs identified in this study. Given that this is a pilot study, we had restricted our comments to the global trends observed with our gene enrichment analyses. However, given that some of the higher expressed miRNAs have been associated with proliferative and invasive features in other pregnancy pathologies, and in cancer, we have made sure to direct the reader’s attention to studies in these other study fields, which further validate that the delivery of some of these miRNAs by placental EVs may lead to some of the clinical features of placenta percreta. Therefore, we have made significant additions to our discussion by as follows:

  • Lines 558 to 582. Although, we are the first, to our knowledge, to determine that the deregulation of these EV-miRNAs may be associated with percreta pregnancies, recent studies have shown that the deregulated expression of miRNAs (i.e., as EV-miRNAs and circulating miRNAs) is also associated with other adverse pregnancy outcomes [80,81], including preeclampsia and spontaneous pre-term birth [82-85]. For example, miR-151, which we found to be upregulated in placental EVs of placenta percreta pregnancies, is known to be abundantly expressed in EVs derived from both trophoblast stem cells (TSCs) and syncytial derived trophoblasts (SynT) and it plays a role in abnormal placentation, which results in endothelial cell hypoxia and the development of preeclampsia [86]. Additionally, miR-486, which we found to be downregulated in placental EVs of placenta percreta pregnancies, has been found to be upregulated in EVs derived from human placental microvascular endothelial cells and is associated with the regulation of proliferation, migration, and invasion of trophoblast cells, thus contributing to poor placentation and the clinical manifestation of preeclampsia [87,88]. Furthermore, in spontaneous pre-term birth, several studies have shown that miR-199a, which we found to be upregulated in placental EVs of placenta percreta pregnancies, plays a crit-ical role in mediating the opposing effects of estrogen and progesterone in uterine contractility during pregnancy, and its downregulation during pregnancy results in spon-taneous preterm birth [89,90]. Although the role of these miRNAs (i.e., miR-151, miR-199a, miR-486, miR-122) has not been explicitly studied in the context of placenta percreta, their identification warrants further analyses on their putative role in the context of this pathology. Considering that our EV-CATCHER assay allows for the purification of intact, functional EVs [91], we propose that PLAP+ EVs purified from maternal plasma from both percreta and previa pregnancies may be tested to treat cells in vitro in order to identify the putative molecular pathways involved in PAS.
  • Lines 583 to 612. “Biologically, the fundamental pathology of PAS involves an overly invasive phenotype of placental cells, which can result in a significant potential for maternal morbidity and mortality [92]. When evaluating the literature for a correlation between the differential expression of these top 6 miRNAs (i.e., miR-486, miR-199a, miR-122, miR-151, miR-378 and miR-340) and cellular invasion, we identified many studies implicating them in the control of cell migration in human cancer cells [93-100]. For example, several stud-ies have demonstrated that miR-486, which we found to be downregulated in placental EVs of placenta percreta pregnancies, is suppressed in different cancer types, including lung, colorectal, and thyroid carcinoma [101-104], but when transfected into cancer cells it leads to a suppression of cell migration, angiogenesis, and invasion [105]. Studies on miR-122, which we found to be downregulated in placental EVs of placenta percreta pregnancies, have shown that when it is delivered by EVs to both colorectal and breast cancer cells, it significantly reduces their metastatic capacity, and thus its decreased ex-pression in cancer cells has been associated with increased cellular invasion [106]. Moreover, miR-199a, which we found to be upregulated in placental EVs of placenta percreta pregnancies, has been shown to contribute to the progression of malignant tumors, specifically when increased in plasma EVs, it promotes cellular proliferation and migration of neuroblastoma cancer cells [107]. Interestingly, all 6 miRNAs have also been associated with the regulation of epithelial-to-mesenchymal transition (EMT), a mechanism common to cancer metastasis [108-111]. These findings are relevant because we have previously shown that EMT promotes cytotrophoblast to extravillous trophoblast (EVT) differentiation and that the EMT signal is enhanced or prolonged in PAS [11]. Particularly, as we determined that miR-122, miR-378 and miR-486 are downregulated in placental EVs in our PAS cases, it is relevant to note that they have also been reported to exert inhibitory actions on EMT in cancer [112-120]. Importantly, miR-151, miR-199a and miR-340, which we found to be upregulated in placental EVs of PAS pregnancies have been shown to both promote and inhibit the EMT process, depen-ing on the cellular context [100, 121-126]. Therefore, the differential expression of EMT-associated placental EV-miRNAs in the context of PAS raises important questions about their putative pathological role during these pregnancies, and thus will warrant further evaluation…”
  • Lines 616 to 622. “…Specifically, our miRNet analysis of the top 14 miRNAs upregulated in EVs that we iso-lated from our PAS show a strong interaction with ZEB1. It is important to note that ZEB1 has been shown to play a pivotal role in enabling proliferation, invasion, and EMT for trophoblast cells during pregnancy [127]. However, the mechanisms by which trophoblast cells achieve these biological effects remain unclear, but we hypothesize that placental EVs transporting these miRNAs may provide a significant stimulus for invasion...”
  • Lines 622 to 631. “…Furthermore, we noted that 25 out of the 40 miRNAs found to be differentially ex-pressed in EVs purified from the plasma of our PAS cases belong to the chromosome 19 miRNA cluster of which 8 belong to the miR-500 miRNA family (i.e., miR-545, miR-501, miR-502, miR-519a-1, miR-519a-2, miR-525, miR-518e and miR-512), which have been found to promote invasion [128]. Altogether, our miRNA NGS analyses identified that circulating PLAP+ EVs contain miRNAs that are differentially expressed in placenta percreta pregnancies (i.e., invasive phenotype), when compared to previa pregnancies, suggesting that circulating placental EVs may contribute to the PAS pathology but importantly that they may provide globally quantifiable biomarkers for the detection of this condition…”

Round 2

Reviewer 1 Report (Previous Reviewer 2)

Comments and Suggestions for Authors

I am satisfied with the author's reply. My concerns have been completely addressed.

This manuscript is a resubmission of an earlier submission. The following is a list of the peer review reports and author responses from that submission.

Round 1

Reviewer 1 Report

Comments and Suggestions for Authors

The authors explored the possibility of identifying soluble/serological predictors of placenta percreta.

I have some concerns about the select group. First, the control group is also a "pathological" group, as placenta previa is a condition characterized by chronically reduced/low fetal blood and oxygen. Also, there are no details on the location of these placentas - are they all central? some marginal? there are many differences.

The authors said they collected 16 samples from placenta percreta. Given the incidence of 1-2/1000 of the entire PAS spectrum (accreta+increta+percreta (line 58), I must infer that the authors skimmed at least 8,000-16,000 placentas: consecutive?. Perecreta is the rarest condition, approximately 5-7% of the PAS spectrum, the screened placentas should have been greater than 220,000. The authors did not provide any time interval during which the sample was collected, nor any clinical details on gestational age, which is collectively defined as 33-35 weeks gestation This means that all pregnancies are preterm, I would strongly consider studying a circulating factor without knowing the exact fetal condition.

No information was provided about the obstetric history of the patients (how many primiparous? multiparous?), their age, their ethnicity, etc.

The molecular part of the study is well described and extremely interesting. However, the selection and description of the population studied cannot be overlooked so easily.

Reviewer 2 Report

Comments and Suggestions for Authors

In this manuscript entitled "Customizing EV-CATCHER to purify placental specific extracellular vesicles from maternal plasma to detect placental pathologies”, the authors used extracellular small vesicles purified to identify potential small-RNA biomarkers for PAS. 

This first description of the purification of extracellular vesicles using EV-Catcher is an interesting contribution to the field, but not enough to justify publication. This description lacks specific quantitative data on purified vesicles. Little data or evidence has been presented that the vesicles are from patients.

Major:

1.     There is little evidence that the plasma obtained contains vesicles. The authors should show that PLAPs and ALPLs can be detected in the obtained plasma. Alternatively, immunolabeling experiments using electron microscopy should be performed to demonstrate the presence of PLAP or ALPL in the vesicles obtained.

2.     The authors should write the amount of plasma used for EV-Catcher and the purified amount. The authors should also analyze whether the purification solution is enriched with PLAP or ALPL.

3.     In Fig.1c, what is the CD? written in red, is it a mistake for PLAP? Are the data in Fig. 1c for a single vesicle? If so, it needs to be quantitatively shown that most vesicles contain similar proteins.

Minor:

4.     In Fig. 1a, the authors should describe how the recombinant proteins used for electrophoresis were obtained.

5.     In line 109, are extracellular vesicles the same as exosomes? The formation process of extracellular vesicles should be explained in the Introduction.

6.     In Fig. 1b and c, the values of the scale bars are unknown and should be written in the legend.

7.     If the experiment involves the use of patient tissue, the use should be approved by the institutional ethics committee. Please write in the approval number.
